# Structural characterization of the full-length Hantaan virus polymerase

Jeremy R. Keown[1,2☯¤]*, Loïc Carrique[1☯], Benjamin E. Nilsson-Payant[3,4,5], Ervin Fodor[6], Jonathan M. Grimes[1]*

1 Division of Structural Biology, Centre for Human Genetics, University of Oxford, Oxford, United Kingdom, 2 School of Life Sciences, University of Warwick, Coventry, United Kingdom, 3 Institute for Experimental Virology, TWINCORE Centre for Experimental and Clinical Infection Research, Hannover, Germany, 4 Cluster of Excellence RESIST (EXC2155), Hannover Medical School, Hannover, Germany, 5 Department of Microbiology, Tumor and Cell Biology, Karolinska Institutet, Stockholm, Sweden, 6 Sir William Dunn School of Pathology, University of Oxford, Oxford, United Kingdom

☯ These authors contributed equally to this work.
¤ Current Address: School of Life Sciences, University of Warwick, Coventry, United Kingdom
* jeremy@strubi.ox.ac.uk (JRK); jonathan.grimes@strubi.ox.ac.uk (JMG)

**Data Availability Statement:** CryoEM data generated in this study have been deposited in the PDB and EMDB with the following accession codes 8P1J/EMD-17351 (HTNV RNA Free- Core), 8P1K/

## Abstract

*Hantaviridae* are a family of segmented negative-sense RNA viruses that contain important human and animal pathogens. *Hantaviridae* contain a viral RNA-dependent RNA polymerase that replicates and transcribes the viral genome. Here we establish the expression and purification of the polymerase from the Old World Hantaan virus and characterise the structure using Cryo-EM. We determine a series of structures at resolutions between 2.7 and 3.3 Å of RNA free polymerase comprising the core, core and endonuclease, and a full-length polymerase. The full-length polymerase structure depicts the location of the cap binding and C-terminal domains which are arranged in a conformation that is incompatible with transcription and in a novel conformation not observed in previous conformations of cap-snatching viral polymerases. We further describe structures with 5′ vRNA promoter in the presence and absence of a nucleotide triphosphate. The nucleotide bound structure mimics a replication pre-initiation complex and the nucleotide stabilises the motif E in a conformation distinct from those previously observed. We observe motif E in four distinct conformations including β-sheet, two helical arrangements, and nucleotide primed arrangement. The insights gained here guide future mechanistic studies of both the transcription and replication activities of the hantavirus polymerase and for the development of therapeutic targets.

## Author summary

Hantaviruses are widely distributed viruses that can undergo zoonosis and cause severe disease in humans. Old World hantaviruses, found predominantly in Asia and Europe, can cause a haemorrhagic fever with renal syndrome. Presently there are no approved specific treatments for infection and current vaccines are of limited efficacy. One of the viral encoded proteins called the polymerase has been identified as a good candidate for therapeutic development owing to its conservation, multiple enzymatic activities, and core

EMD-17352 (HTNV RNA Free- Core and endonuclease), 8P1L/EMD-17353 (HTNV RNA Free- Full-length), 8P1M/EMD-17354 (HTNV 5′ vRNA), and 8P1N/EMD-17355 (HTNV 5′ vRNA and ATP).

**Funding:** This work was supported by Wellcome Investigator Awards (200835/Z/16/Z and 222510/Z/21/Z to JMG), a University of Oxford-Medical Sciences Division Pump Priming Award (0012329 to JRK), a Medical Research Council programme grant (MR/R009945/1 to E.F.), the Deutsche Forschungsgemeinschaft (DFG, German Research Foundation) under Germany's Excellence Strategy - EXC 2155 (390874280 to BEN) and the Swedish Research Council (2023-02595 to BEN). The views expressed are those of the author(s) and not necessarily those of the NHS, the NIHR, or the Department of Health. The funders had no role in study design, data collection and analysis, decision to publish, or preparation of the manuscript.

**Competing interests:** The authors have declared that no competing interests exist.

function in the replication and transcription of the viral genome. Here we developed protocols to express highly pure polymerase and subsequently determined its three-dimensional structure in RNA-free, RNA-bound, and RNA/NTP bound states. We observe a full-length polymerase structure with the mobile domains arranged in a location not previously observed in either hantavirus polymerases or in polymerases from the wider bunyavirus family. Comparison of the polymerase core showed extensive rearrangement of a region called motif E and we observe the region in helical, sheet, and loop conformations. Using these models we are able to propose a scheme for the replication initiation. These findings are important in developing potential therapeutic inhibitors specific for hantaviruses.

## Introduction

The *Hantaviridae* are a family of segmented negative-sense RNA viruses (sNSV) from the larger class of the *Bunyaviricetes* [1]. The family can be further separated into the Old World viruses (for example Hantaan virus, HTNV) and New World viruses (for example Andes virus). Unlike most members of the *Bunyaviricetes* which are spread by arthropods, *Hantaviridae* are transmitted by rodent vectors [2]. Hantaan virus (HTNV) is an Old World Hantavirus that is found in China, Russia, and South Korea [2]. The virus is spread by *Apodemus agrarius* (striped field mouse) and causes haemorrhagic fever with renal syndrome with a case fatality rate of 1–15% [3,4]. The virus was the cause of many severe illnesses during the Korean war [4].

*Hantaviridae* have three genome segments termed the small (S), medium (M), and large (L) [5]. The S and M segments encode the viral nucleoprotein and surface glycoproteins Gn/Gc, respectively [6]. The L segment encodes the L protein, an approximately 250 kDa polymerase. The polymerase is a multifunctional enzyme that contains an N-terminal His+ endonuclease [7], a RNA-dependent RNA polymerase (RdRp), and a presumed cap binding domain (CBD) at the C-terminus [8]. The viral polymerase carries out both transcription (production of viral mRNA) and replication (production of genomic RNA). A single copy of the polymerase, many copies of the nucleoprotein, and a single piece of viral RNA collectively assemble the ribonucleoprotein (RNP) complex [6]. In these RNP complexes the viral RNA is wound around a nucleoprotein scaffold, while both the 5′ and 3′ termini of the RNA are bound by the viral polymerase.

In transcription, the viral polymerase interacts with cytoplasmic host capped mRNA via a poorly understood mechanism, subsequently cleaving a short capped primer in a process known as cap-snatching [7]. This primer is used to initiate transcription. There is evidence that transcription utilises a prime-realign style mechanism [9] and that the transcripts from the M subunit are polyadenylated while L and S are terminated with a structured RNA [10]. It has been suggested that translation happens in a coordinated manner with transcription in bunyaviruses [11]. Replication is a two step process, where the viral RNA is first copied to a positive sense complimentary intermediate before this is used as a template to produce identical copies of the original negative sense viral RNA [6]. Replication initiation has been reported to occur from position 3 where it is first extended by three nucleotides before realignment of the primer to position -1 creating a product with an extra G at the 5′ end [9].

The structures of viral polymerases from a limited set of *Bunyaviricetes* have recently been determined including those from Dabie bandavirus (DBV, previously called Severe Fever with Thrombocytopenia Syndrome virus) (*Phenuiviridae* family) [12–14], Lassa and Machupo

viruses (*Arenaviridae* family) [15–18], Rift Valley fever virus (RVFV, *Phlebovirus* family) [19], La crosse virus (LACV, *Peribunyaviridae* family) [20–22] and the recent hantavirus polymerase [23,24]. These structures have detailed many aspects of both replication and transcription for these viruses in complex with RNA and accessory viral proteins (reviewed recently in [8]). The viral polymerase from *Bunyaviricetes* viruses is an attractive target for the development of antiviral therapeutics given its conserved function within and perhaps between families. To aid development of these therapeutics the structure of the full-length viral polymerase is required in high resolution.

In this work, we present the full-length structure of the Old World HTNV polymerase revealing in molecular detail the structural organisation of the previously unobserved flexible endonuclease and C-terminal domains. Comparison of our models to those previously determined for other bunyaviricetes reveals an arrangement of these domains not previously observed. Determination of structures without RNA, in the presence of 5′ vRNA promoter, or in the presence of 5′ vRNA promoter and nucleotide reveals significant rearrangement of the motif E and a non-canonical coordination of the active site catalytic magnesium, whilst the nucleotide bound polymerase mimics a NTP primed pre-replication initiation state.

## Results/Discussion

### Purification and initial characterisation of the Hantaan virus *L protein*

The gene encoding the full length Hantaan virus L protein with an N-terminal OctaHis tag, a twinStrep, and a TEV-protease site was synthesised and cloned into the pFastbac vector for expression in SF9 insect cells. Previous studies have shown that polymerases of the *Hantaviridae* have a highly active endonuclease which effectively suppress protein expression including of the viral polymerase [25]. We therefore carried out initial expression experiments with both the wildtype (wt) polymerase and a polymerase with a previously characterised endonuclease mutation (D97A) which was previously shown to inhibit endonuclease activity [6,7,26]. Here, we did not observe differences in the expression, purification, or stability of both proteins (S1A Fig). We validated that the D97A mutant was unable to cleave an RNA substrate in the presence of magnesium, manganese, or promoter RNA whereas the wt protein contained potent nuclease activity at micromolar concentrations of manganese (S1B–S1D Fig).

### Structure of the HTNV polymerase core

We first collected single particle cryogenic electron microscopy (cryo-EM) data of the viral polymerase in the absence of viral RNA promoters. We observed multiple views of the polymerase and were able to reconstruct maps to high resolution (S2A Fig). From this dataset, we were able to determine the structure of the polymerase in two conformations comprising either only the polymerase core (residues 226–1600) or with the core and the endonuclease ordered (residues 1–1600) to resolutions of 2.8 Å and 3.24 Å, respectively (Fig 1A and 1B). In these structures, the C-terminal domains were not observed.

At the N-terminus of the polymerase we observe the endonuclease domain (residues 1–215) (Fig 1A). Linking the endonuclease to the core lobe (215–261) is a 46 amino acids linker that runs along the fingers and palm domain (Fig 1A). The core lobe contains the 5′ RNA termini binding site. Two loops in this lobe (392–402 and 515–523) are not ordered in the model and these likely contact the 5′ RNA. Residues 392–402 appear analogous to the PA-arch of the influenza virus polymerase. The core lobe also contains motif G (627-RY) that forms part of the polymerase active site. C-terminal to the core lobe are the fingers and palm domains which contain the remaining motifs A-F (Fig 1B).

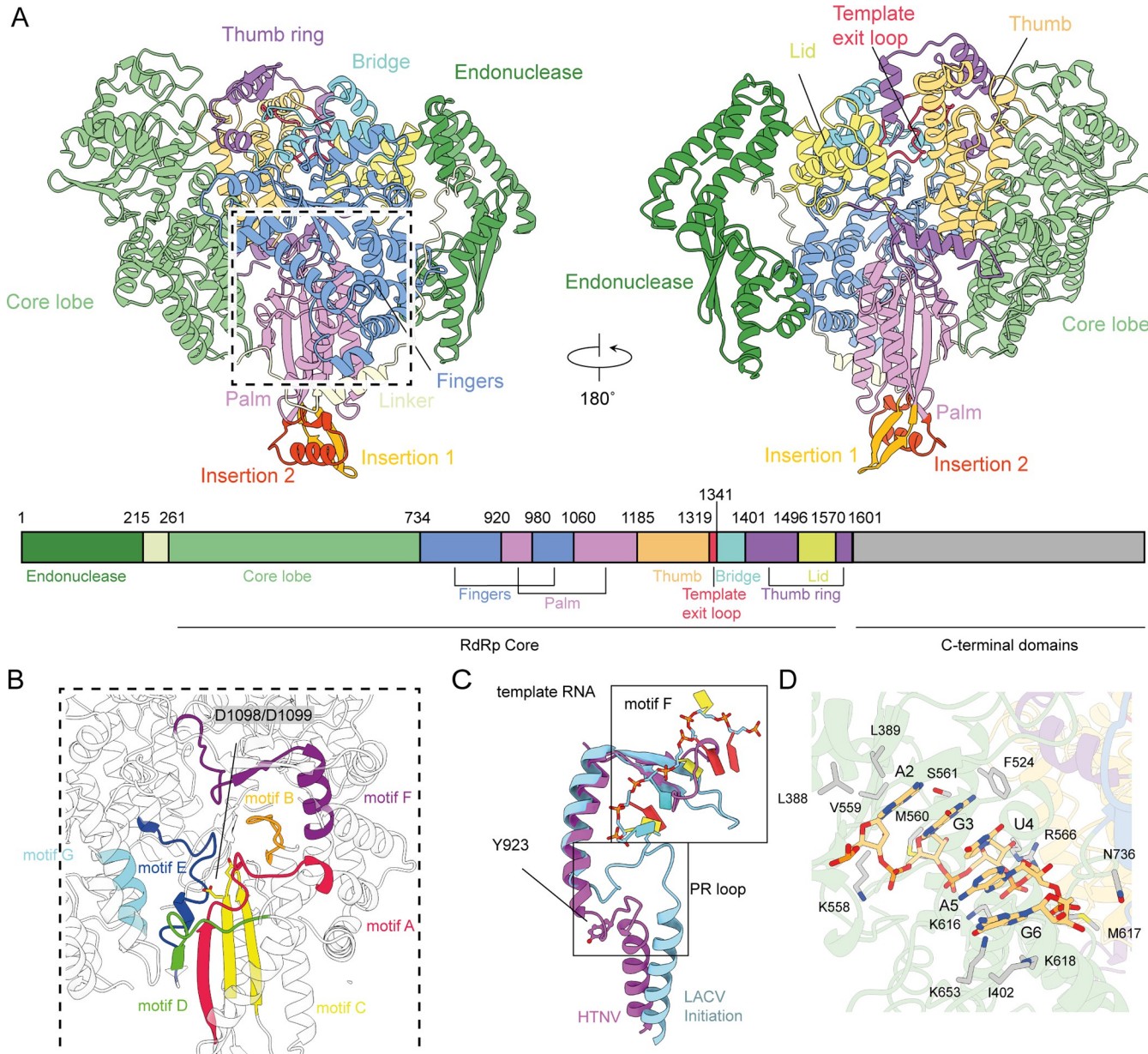

**Fig 1. The endonuclease and core HTNV polymerase structure.** (A) Two orientations of the HTNV polymerase endonuclease and core model with the domains and regions annotated. Domain scheme is shown in linear form below. (B) Motif annotation of the polymerase core shown in the same orientation as in (A). The active site aspartate residues are annotated and shown in atom representation. (C) Comparison of the HTNV RNA free polymerase (purple) and the LACV polymerase (PDBID: 7ORN), showing how motif F blocks entry of the RNA template in the HTNV structure. (D) The binding of the bases 2–6 of the 5′ promoter RNA and interacting side chains are shown with atom representation and annotated.

Motif A in the palm domain contains the conserved sequence 972-Dx$_2$KW-976 where the aspartate residue participates in coordination of an active site divalent cation. Motif B with the motif 1058-QGx$_5$SS-1066 has a role in binding both the incoming nucleotide triphosphate (NTP) and template strand. Found at the tip of two β-strands, motif C contains the catalytic aspartic acids 1097-SDD-1099 which coordinate the active site divalent cations. Motif D contributes two conserved lysines 1160-KK that line the NTP entry channel and form part of the scaffold for the palm domain. Motif E is located in a hinge region between the palm and

thumb domains and contains the consensus sequence 1170-Ex$_2$S-1173. Motif F is found in the fingers domain with the sequence 884-KxQx$_5$R-892. . .902-Rx$_6$E-909 (Fig 1C). In our RNA free model motif F is positioned such that it would block the RNA template entry channel (Fig 1C).

We observe two insertions in the palm domain, insertion 1 (residues 948–963) and insertion 2 (residues 1109–1135) (Fig 1A), as reported in recent similar studies [23,24]. Insertion 1 forms an additional short β-strand on to the palm and extending motif A. Insertion 2 forms a short helix that packs against insertion 1. Insertions have been observed in the palm domain of negative sense RdRp but they are not structurally similar to those observed here (S3A Fig) [20].

Linking the finger and palm domains is a stretch of residues termed the prime-realign (PR) loop [21]. The PR loop has a consensus sequence in *Peribunyaviridae* of DEMIS, while in the *Hantaviridae* the sequence is EEYIS. In our RNA-free HTNV structures we observed the loop 921-EEYIS-925 in a retracted conformation (Figs 1C and S3B). This retracted conformation is similar to that observed for LACV polymerase pre-initiation conformation, RNA-free DBV, or RNA-free Lassa virus polymerase structures [12,15,22]. In the LACV initiation conformation the loop rearranges and packs against the 3' end of the vRNA (Fig 1C and S3B) [21].

After extensive screening we were able to prepare a polymerase complex in which bases from the 5' promoter RNA could be observed in the RNA binding site (S2B Fig). This complex contained 3'/5' promoter RNA, capped RNA, UTP, and AMP-PNP where the reaction was allowed to proceed at 30 ˚C for 45 minutes. The sample was modified using NHS-PEG$_4$, a process referred as PEGylation before grid preparation [27]. In this complex we observe only a short piece of RNA corresponding to bases 2–6 of the 17 nucleotide long 5' vRNA promoter (Figs 1D and S3C). The RNA is bound in the core lobe of the polymerase, in a position analogous to the hook position seen in HTNV polymerase and other negative strand RNA viruses (S3D and S3E Fig). From our data we are unable to conclude where the first base is located or what interactions may form within the RNA of a potential hook type structure. Why we only observe these five bases and not the full 5' vRNA is not clear.

The binding of even this partial 5' vRNA is enough to cause significant changes to the RdRp core. Comparing the RNA bound structures with the RNA free models shows a straightening of motif F when RNA is bound (S3F Fig). In the bent RNA free conformation, motif F contacts the thumb ring region while with RNA bound new contacts are formed with the helix of motif G (residues 623–631). The RNA bound conformation we observe here is similar to that previously observed for LACV polymerase (S3G Fig) [20–22]. In our RNA free structures motif F is orientated such that the template entry channel is blocked, whilst binding of the 5' promoter RNA rearranges the motif, opening the channel (Fig 1C).

The structural features and rearrangements observed upon the addition of 5' vRNA promoter to the polymerase core are conserved between our model presented here and those recently reported with an extended section of 5' vRNA [23].

## The structure of the HTNV polymerase C-terminal domains

During our attempts to generate an RNA bound polymerase complex we observed a subset of particles in which the full-length polymerase had become ordered. This complex was formed by addition of 5' and 3' promoter vRNA, a capped RNA primer, and NTPs and incubated on ice for 1 hour before pegylation with NHS-PEG$_4$. The full-length polymerase structure determined from this dataset did not contain RNA, while a small subset of the dataset contained a 5' RNA promoter and core polymerase structure identical to that described above. The full-length structure was determined to a resolution of 3.32 Å (S4A Fig).

The HTNV polymerase C-terminal region contains a bipartite midlink domain spanning residues 1601–1704 and 1825–1920 (Fig 2A and 2B). The midlink domain has extensive contacts with the N-terminal endonuclease domain, the CBD, and the C-terminal domain and likely serves as the pivot point as the CBD and C-terminal domains undergo rearrangement.

The CBD (residues 1704–1825) is formed by five stranded β-sheet with a five-turn helix packing against the solvent exposed face, while the other face packs against the thumb ring domain of the polymerase core. Extended from the CBD is a short four stranded β-sheet. Analysis of the CBD suggests a putative binding site comprising residues Y1710 and Y1725 which would sandwich the cap of the cleaved mRNA (Fig 2A inset). A recent study of Sin Nombre virus also suggested these residues would be important in cap binding [24]. The two residues are separated by approximately 8 Å, a similar distance observed between cap binding domain residues in other cap-snatching polymerases. Comparison of the HTNV CBD to the DBV CBD shows a striking similarity with a similar sized central β-sheet, a helix packing against one face, and a small β-sheet extension (Fig 2C). Comparison to the CBD from the LACV and influenza virus shows a similar but larger central β-sheet but no short sheet extension and additional structural complexities (Fig 2C). Previous work [28] has suggested the vial nucleoprotein may act as the cap binding protein in the context of the replication complex; our results suggest this function is contained in the viral polymerase.

The C-terminal domain is a small helical domain (residues 1921–2151) that contains a disordered region between residues 1947–2027 (Fig 2A inset). An extensive structure homology search of the C-terminal domain using the Foldseek [29] server found no proteins with a related fold. By analogy to the DBV C-terminal domain the disordered residues after the C-terminal domain may form the Lariat domain, that has been reported to interact with the endonuclease domain in DBV [12,14]. The close association of the endonuclease and the C-terminal domain would make this interaction possible.

## The HTNV full-length polymerase conformation

The global conformation of the polymerase is conserved between the core + endonuclease and full-length structures; however, we note some interesting differences. Outside of the RdRp active site the thumb ring (residues 1455–1485) region is now fully ordered in comparison to the core + endonuclease model.

Within the polymerase core, residues 1320–1340 are observed in two different ordered conformations and are disordered in one model. In the core + endonuclease model these residues form a plug between the bridge and thumb domains blocking the putative exit channel (Fig 3A inset). In the full-length model these residues form an open loop but the channel does not appear to be sufficiently open to allow the passage of RNA out of the polymerase. In the HTNV elongation conformation these residues have undergone complete rearrangement opening the channel and allowing a clear path for the RNA to exit (Fig 3A, inset). In the presence of the 5′ vRNA, residues 1319–1328 are disordered, fully opening the channel (Fig 3A, inset). Previously these residues have been termed the priming loop [23,24], however given their distance from the active site and their described role here we will refer to them as the template exit loop.

In the full-length conformation, we observe the endonuclease active site oriented towards the polymerase active site and the putative CBD binding site oriented away from the polymerase (Figs 2A and 3A). In this structure, as was observed for the RNA-free core+endonuclease conformation, the template entry channel is occluded by motif F (residues 885–909) (Fig 1C).

Comparison of the endonuclease and C-terminal domain arrangement to those conformations previously determined for members for the *Bunyaviricetes* class reveals this conformation

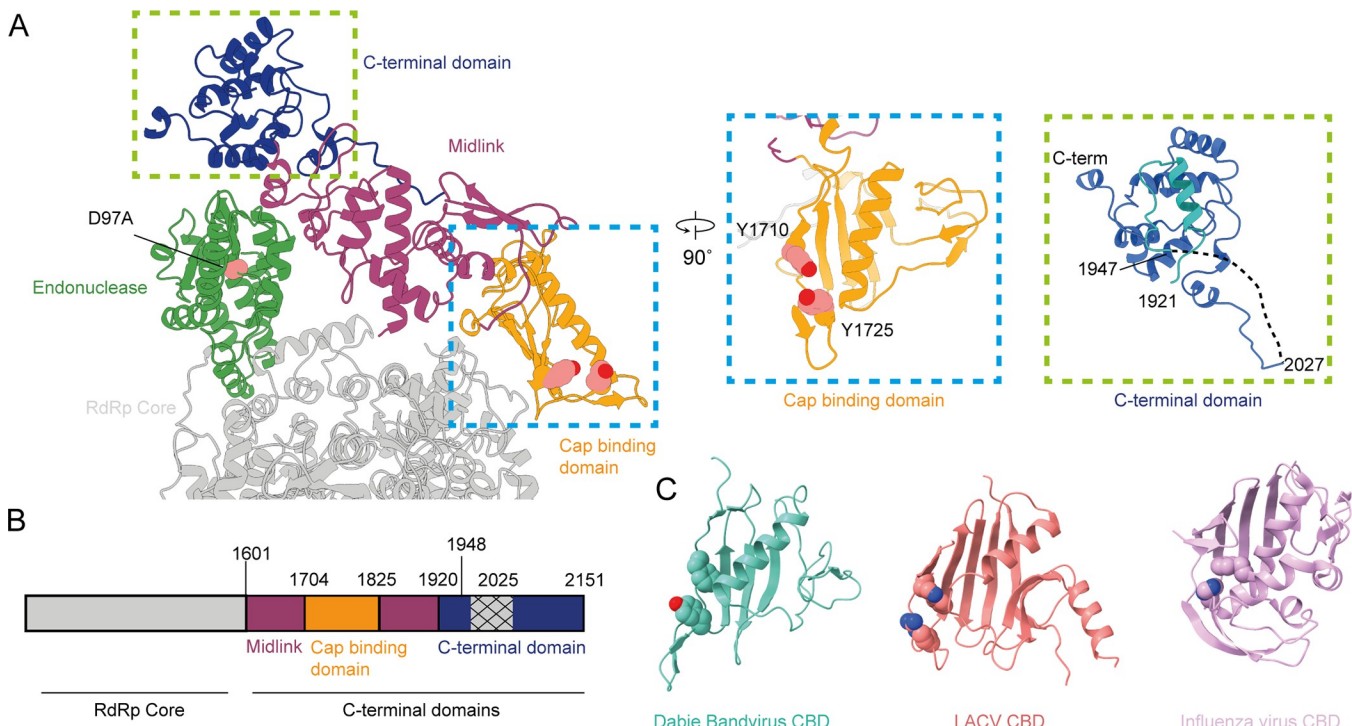

**Fig 2. The C-terminal domains of HTNV polymerase.** (A)The relative positions of the C-terminal domains from the full-length polymerase are show with key active site residues annotated. Inset images show a 90˚ rotation and zoom of the cap binding and C-terminal domains. (B) Scheme detailing the domain boundaries for HTNV polymerase. Hatched region indicates residues not observed in our model. (C) CBD for the viral polymerase from DBV (PDBID: 7ALP), LACV (PDBID: 7ORJ), and Influenza virus (PDBID: 6RR7).

to be distinct (Fig 3B). Structures including the flexible domains representing pre-initiation, cap cleavage, and elongation conformations have all been reported in the literature with a characteristic position of the flexible domains reported for each [8]. Recent structures reporting the elongation and preinitiation conformations of the HTNV polymerase did not observe the endonuclease or C-terminal domains [23].

## Rearrangement of the polymerase active site upon RNA and nucleotide binding

An RNA and nucleotide bound sample was generated by mixing 5′/3′ vRNA promoters, capped RNA, NTPs, magnesium, and manganese prior to grid preparation (S4B Fig). Analysis of this sample by cryoEM revealed a polymerase core with 5′ vRNA promoter bound as in the previous structure and with a single nucleotide bound in the active site. The density for the three phosphates is strong, while the density for the ribose and base is weaker (S5A Fig). The positions of the NTP and magnesium are similar to those observed in recent bunyavirus polymerase structures [13]. We have modelled the nucleotide base as an adenosine, due to the size of the sidechain density but it is reasonable to suspect it may be a mixture of all four nucleotides. In the polymerase active site the NTP is coordinated by the catalytic magnesium ions which we could model from the density (Figs 4A and S5A), which are in turn coordinated by the active site residues D1098/1099, D972, and E1177. While the aspartate residues participate in canonical interactions with the active site magnesium ions, E1177 is from motif E. In contrast to the two magnesium ions observed in our model a single ion was observed in a recent study [23].

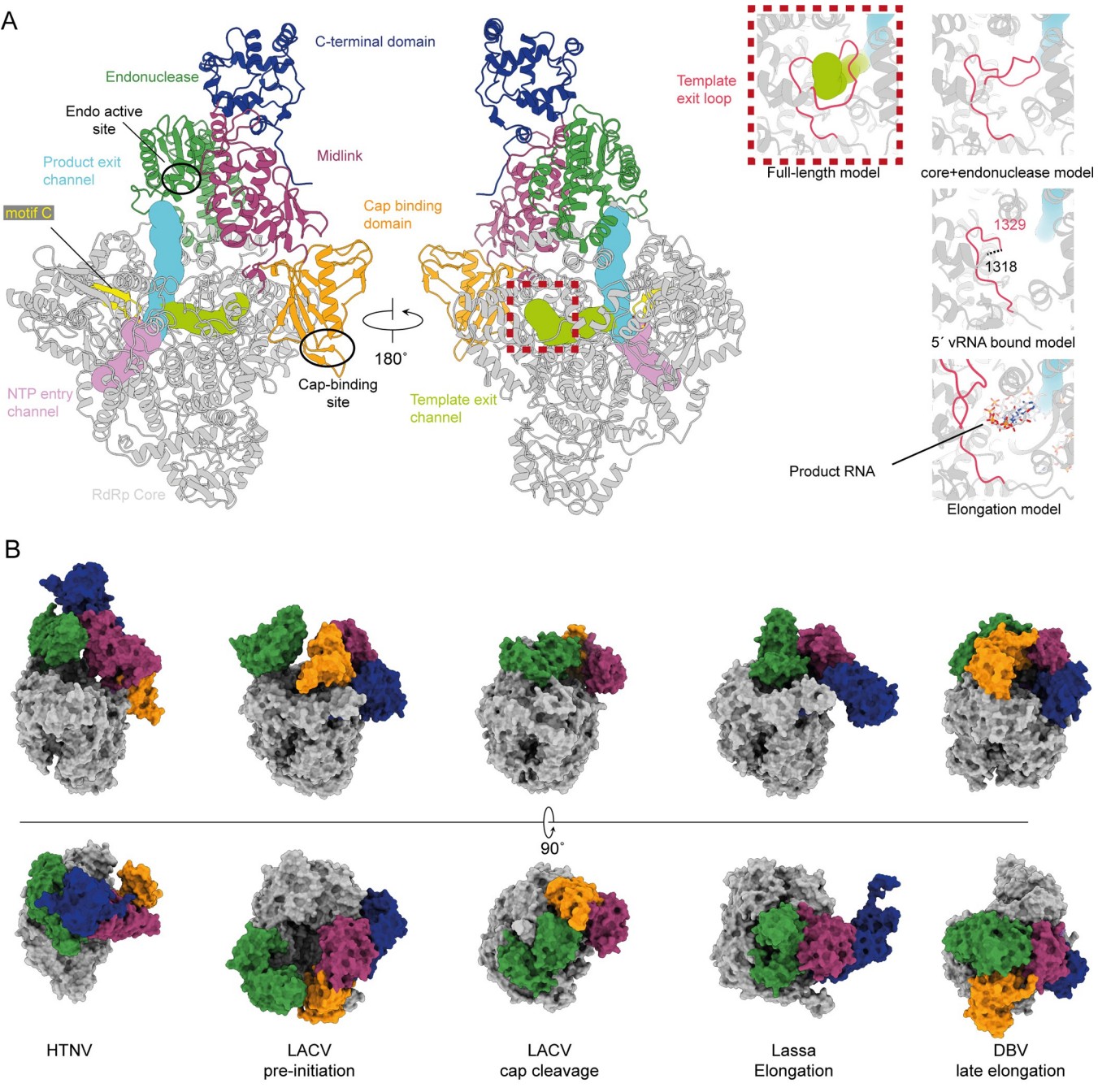

**Fig 3. The global conformation of the HTNV full length polymerase.** (A)The full-length polymerase conformation is shown with flexible domains and the polymerase active site (motif C) annotated. The endonuclease and CBD sites have been highlighted. Inset images show the arrangement of the template exit loop (residues 1320–1340) in the full-length model (dashed box), core + endonuclease, 5′ vRNA bound, and elongation (PDBID: 8C4V) models. Tunnels within the polymerase core are shown and annotated. (B) The relative domain positions of the flexible domains are shown for previously observed polymerase structures. LACV pre-initiation (PDBID: 7ORJ), LACV cap cleavage (PDBID: 8ASD), Lassa virus elongation (PDBID: 6ZB6) and DBV late elongation (PDBID: 7ORN) polymerase structures.

In the presence of only the 5′ vRNA we observe motif E (residues 1167–1185) forming a three stranded β-sheet (Figs 4B and S5C). This arrangement is analogous to that recently observed with HTNV polymerase in complex with a 25 nucleotide 5′ vRNA promoter [23]. Binding of magnesium and NTP to the active site causes a complete rearrangement of motif E into a nucleotide primed conformation, reorienting E1177 into the active site such that it now

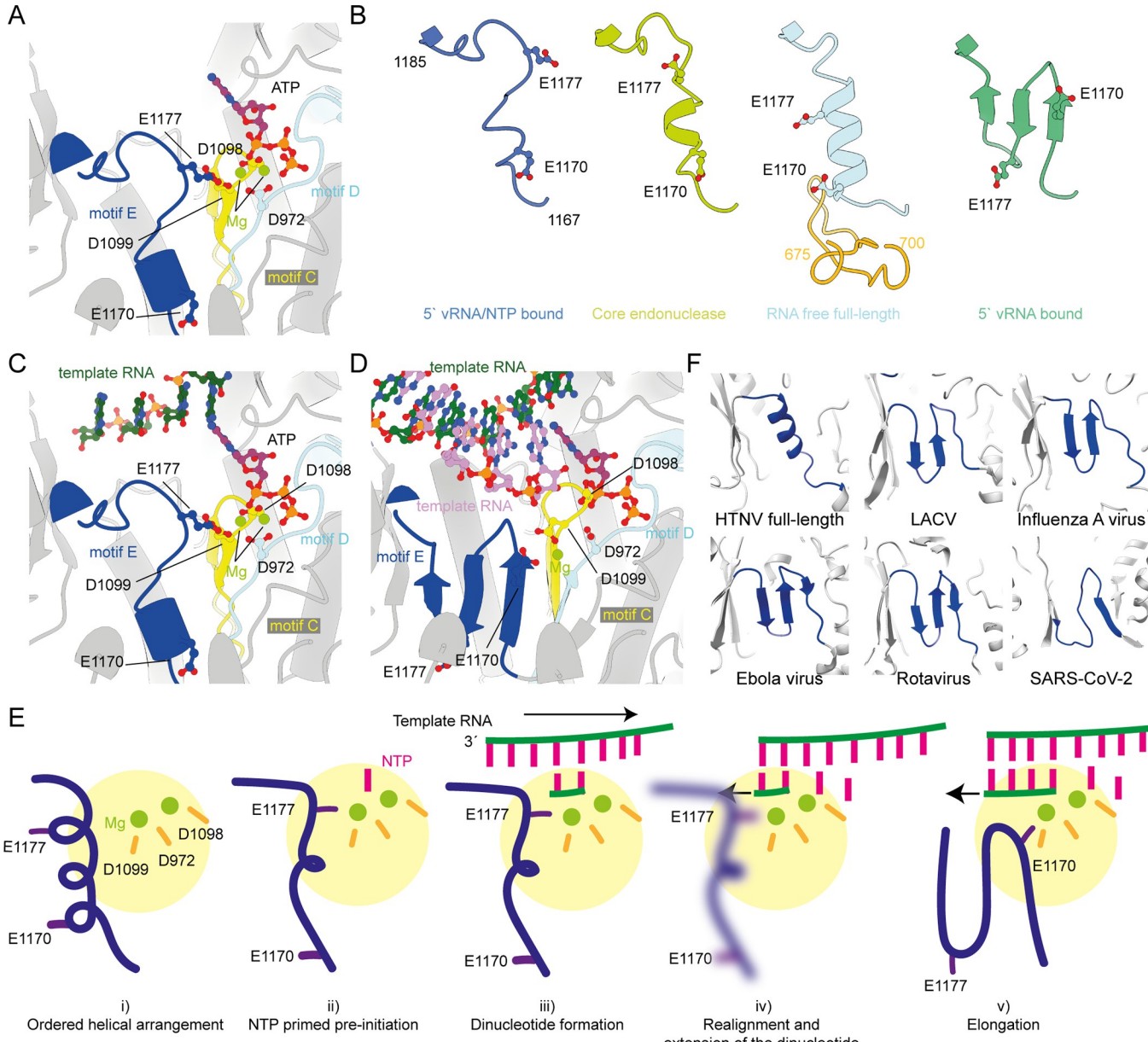

**Fig 4. Nucleotide binding causes motif E to change conformation.** (A) Coordination of ATP in the active site with motif C, D, and E and the residues which coordinate the magnesium ions shows in ball and stick representations. (B) Four HTNV polymerase were aligned and the residues from motif E (1167–1185) were extracted to show their different arrangements. The position of E1177 is show in stick orientation. For the RNA free full-length model residues 675–700, which are ordered in the full-length model, are also shown. (C) Modelling of the HTNV pre-initiation RNA (PDBID: 8C4U) demonstrates that the 4th base from the 3′ end of the template would be correctly positioned above the NTP in the active site. (D) Alignment of the NTP into the HTNV elongation model (PDBID: 8C4V). (E) Proposed model describing the movement of RNA and motif E during the early stages of replication. (F) A zoom of motif E in polymerase from HTNV full length, La Crosse (PDBID: 6Z6B), influenza virus (PDBID: 6RR7), Ebola virus (PDBID: 7YER), Rotavirus (PDBID: 2R7Q), and SARS-CoV-2 (PDBID: 6YYT) showing a conservation of a β-strand arrangement.

coordinates one of the two active site magnesium ions (Fig 4A and 4B). This arrangement is in distinct contrast to that observed in the core + endonuclease conformation (Figs 4B and S5C) where these residues form a short helix, as was observed for the HTNV polymerase [23]. In the RNA free full-length structure, residues 675–700 have become ordered, forming a two stranded antiparallel β-sheet which packs against the N-terminal end of motif E, stabilising it into a longer and more rigid helix that is further retracted from the active site (Figs 4B and S5C). Across these different arrangements of motif E, E1177 only enters the active site to coordinate a magnesium ion in the presence of RNA, magnesium, and NTP (Fig 4B).

The 5′ vRNA bound β-sheet conformation of motif E is common to all other bunyavirus RdRp that we could identify in the literature, and we could find no examples of a helix or helix-like secondary structure at this position [8]. Structural comparison of HTNV with 5′ vRNA showed strong conservation to the recently determined Sin Nombre polymerase core structure with 5′ vRNA bound (S6 Fig).

The recent HTNV RNA free polymerase core structure has motif E in the same helical arrangement that we observe in our full-length structure [23]. The elongation and pre-initiation structures reported elsewhere contain the motif E in the three stranded β-sheet arrangement we observe here (Fig 4B). The position of the NTP, magnesium ions, and active site aspartate residues in our model are consistent with previous observations of NTPs in sNSV RdRp active sites (S5B Fig). Modelling of the RNA from the HTNV pre-initiation structure shows the template would be positioned with the 4th nucleotide from the 3′ end aligned to the nucleotide we observe in the active site (Fig 4C). Overlay of the nucleotide into the HTNV elongation model demonstrates how the nucleotide is correctly positioned for addition to the RNA product (Fig 4D). Due to steric hinderance extension of the new product RNA by 1–2 nucleotides would not be able to occur without motif E undergoing a rearrangement into the three stranded β-sheet conformation, as observed in the elongation models (Fig 4E). The helical arrangement of motif E would not be compatible with transcription initiation as it would block the 3′ end of the capped RNA primer entering the active site.

Comparison of the RNA bound polymerase in the presence or absence of NTP reveals the template exit channel loop has rearranged upon addition of NTP such that it is retracting but the channel is still closed (Fig 3A, inset). Though the NTP bound complex was formed with the addition of a capped primer, designed to mimic transcription, given the positioning of the nucleotide we believe this conformation is likely a NTP primed pre-replication initiation state.

## Conclusion

The viral L protein from negative-sense RNA viruses is central to both transcription and replication of the viral genome. Here, we present the full-length structure of the Hantaan virus L protein in a conformation not previously observed and in complex with short promoter RNA and an NTP.

An interesting observation is that motif E can switch between β-sheet, two different helical, and a nucleotide primed conformation. The rearrangement of the motif appears to be in response to an ordering of the polymerase, or addition of nucleotide and/or RNA. Motif E, also known as the "primer grip", is canonically a β-hairpin at the junction of the palm and thumb domains. Comparison of the motif E from segmented or non-segmented negative sense RNA viruses and single or double stranded RNA viruses shows a broad conservation of the motif as a β-strand containing motif (Fig 4F). This is in good agreement with previous structural alignments describing strong conservation of the β-hairpin across diverse virus species, even with low sequence conservation [30,31]. We have been unable to identify a helical motif E in any experimental structure of viral polymerases. Interestingly, modelling of the

HTNV polymerase using Alphafold2 predicts a short helix for motif E, though this prediction is much lower confidence than other regions of the polymerase core.

Using the recent structures of the HTNV polymerase in pre-initiation and elongation states we propose a molecular model for the internal initiation of HTNV genome replication (Fig 4E) [23]. In this model the ordered helical arrangement of motif E serves as a decision point from which either transcription or replication can proceed (Fig 4Ei). Once an NTP enters the active site (Fig 4Eii) motif E rearranges such that the E1177 coordinates a magnesium and blocks entry of a capped RNA primer. The 3′ vRNA can then enter active site where a dinucleotide is then formed internally, (Fig 4Eiii) but further extension is blocked in the nucleotide primed conformation. Upon realignment of the 3′ vRNA, terminal initiation proceeds disrupting the nucleotide primed arrangement of motif E (Fig 4Eiv) into the β-strand arrangement and allowing elongation to progress (Fig 4Ev). Recent structural data suggests the 3′ vRNA can overshoot the active site by four nucleotides [23], with functional data suggesting dinucleotide formation at position 2/3 [23] while previous functional data suggest replication occurs from position 3 to generate an additional G at the 5′ end [9]. In our model we do not propose a specific position for internal initiation, or even if there is one, only that a dinucleotide is the product prior to realignment. In our model the β-strand conformation is a low energy conformation which once formed will require a resetting process to the helical arrangement.

Unlike some negative sense RNA viruses, the *Hantaviridae* utilise internal initiation coupled to a prime-realign mechanism when copying both the negative and positive sense RNA templates. This is in contrast to the influenza viruses which utilise internal initiation with prime-realign only on the positive sense template [32].

Comparison of the full-length HTNV polymerase conformation to influenza virus polymerase conformations for which cap-snatching, transcription initiation, elongation, termination, replicase, and encapsidase conformations have been determined [33,34] does not reveal similarly positioned domains. If instead we consider not the specific domain position, but the relevance of each domain position, we can propose possible roles for this conformation. The full-length conformation captured here likely represents a newly synthesized polymerase that is in a conformation ready to bind newly synthesized RNA. This role is similar to that of the encapsidating influenza virus polymerase observed during replication [35]. This is consistent with an absence of RNA in the structure and closed template entry channel. If we ignore the absence of RNA, it may be that this conformation represents 1) a pre cap-snatching conformation, where the cap is exposed to the solvent or 2) a pre-replication initiation conformation, where the helical motif E is poised for magnesium and nucleotide binding. Future experiments are required to conclusively determine the precise function of this arrangement.

The observations and analysis we have carried out on the HTNV polymerase reveal a complex and mobile enzyme. We describe the previously unobserved C-terminal domains and how these domains arrange in relation to the endonuclease and polymerase core domains. The observation of an NTP in the polymerase active site expands the complex role of motif E in the polymerase activities, however whether the rearrangement of the motif is specific to the *Hantaviridae* family or a general feature of the *Bunyaviricetes class* is not yet clear.

## Materials and methods

### Protein cloning and purification

The *Hantaan virus* polymerase (NCBI Reference Sequence: NP_941982.1) with an N-terminal twin-strep, octaHis-tag and TEV-protease site was synthesised (SynBio) in a pFastbac plasmid. A baculovirus for recombinant protein expression was generated using standard protocols and sf9 insect cells were maintained in Sf-900 II serum-free media (Gibco). 72 hours post infection,

cells were harvested and resuspended in a wash buffer containing 50 mM HEPES, pH 7.5, 500 mM NaCl, 0.05% (w/v) n-Octyl beta-D-thioglucopyranoside, 2 mM dithiothreitol, 10% (v/v) glycerol, one protease inhibitor tablet (Sigma), 5 mg RNAse, and 2.4 mL of BioLock (IBA). Cells were lysed by sonication and clarified with centrifugation. The supernatant was then applied to Strep-Tactin Superflow high capacity (IBA) resin and incubated for 2 hours. The resin was washed with 50 column volumes of a buffer containing 50 mM HEPES, pH 7.5, 500mM NaCl, 0.05% (w/v) n-Octyl beta-D-thioglucopyranoside, 2 mM dithiothreitol, and 5% (v/v) glycerol. Prior to elution the resin was washed with 10 column volumes of 50 mM HEPES, pH 7.5, 500mM NaCl, 2mM dithiothreitol (DTT). The polymerase was eluted with 20 mM HEPES, pH7.6, 500 mM NaCl, 1 mM dithiothreitol, 5% (v/v) glycerol, and 50 mM Biotin. The sample was then concentrated to 0.5 mg/mL.

Mutations were inserted into the wt plasmid using standard cloning techniques. Mutant protein was expressed and purified as for the wildtype.

## Endonuclease activity assays

A Cy5-labelled 27-mer ssRNA substrate (5′-[CY5]GAUGAUGCUAUCACCGCGCUCGUC GUC-3′), a 17mer 5′ vRNA (5′-pUAGUAGUAGUAUGCUCC-3′) and a 19-mer 3′ vRNA (5′-pUUUAGGGAGUCUACUACUA-3′) promoter template were chemically synthesised (Integrated DNA Technologies). Endonuclease activity assays were performed by incubating 1 μg L protein with 100 nM Cy5-27mer ssRNA in 20 mM HEPES, pH 7.5, 150 mM NaCl, 1 mM DTT, 2 U/μl RNaseOUT (Invitrogen) and as indicated 2 μM 5′ vRNA, 2 μM 3′ vRNA, 5 mM MnCl$_2$, and 5 mM MgCl$_2$. Endonuclease assays were incubated for 2 hours at 30°C. Reactions were stopped by adding RNA loading buffer for a final concentration of 45% formamide and 5 mM Ethylenediaminetetraacetic acid (EDTA) and heating samples to 95°C for 5 minutes. Reaction products were resolved by 7 M Urea 20% polyacrylamide Tris-borate-EDTA (TBE) gel electrophoresis (PAGE) in 0.5X TBE buffer. Fluorescence signals were detected in the gel on a Fusion FX7 imager (Vilber). Intensities of fluorescent signals were quantified using the ImageJ software [36].

## Cryo-EM sample preparation

To inhibit RNA degradation, the D97A mutant was used for all cryo-EM samples. During the development of the project, we continued optimisation of reaction conditions to achieve a sample which was homogenous for single particle cryo-EM analysis. The first dataset, which generated the RNA free core and core endonuclease structures was prepared directly from eluted protein without the addition of any components.

The second dataset, that generated the full-length structure, was prepared by mixing 2 μM polymerase, 4 μM 5′ vRNA (5′-UAGUAGUAGUAUGCUCC-3′), and MgCl$_2$ to a final concentration of 5 mM for 10 minutes on ice. To this complex 4 μM 3′ vRNA (5′-UUUAGGGA GUCUACUACUA-3′), 10 μM 12mer capped RNA (5′ m$^7$GpppAmAUCUAUAAUAG 3′), and nucleotide triphosphates to (ATP, GTP, UTP, CTP) a final concertation of 1 mM and incubated on ice for a further 30 minutes. The sample was then pegylated for 15 minutes with 2 mM MS(PEG)4 Methyl-PEG-NHS-Ester (ThermoFisher Scientific) before grids. The sample was incubated on ice to maintain sample stability and enhance RNA binding.

Dataset 3 which generated the 5′ only RNA dataset was prepared by combining 2 μM polymerase, 5 mM MgCl$_2$, 5 mM MnCl$_2$, 4 μM 5′ vRNA (5′-UAGUAGUAGUAUGCUCC-3′), 3′ vRNA (5′-UUUAGGGAGUCUACUACUA-3′), 10 μM 12mer capped RNA (5′ m$^7$GpppAmA UCUAUAAUAG 3′), 1 mM UTP, 1 mM nonhydrolyzable AMP-PNP, and 2 mM MS(PEG)4

Methyl-PEG-NHS-Ester (ThermoFisher Scientific) at 30 ˚C for 45 minutes. Grids were prepared without additional modification.

Dataset 4 which yielded the nucleotide bound structure was prepared by combining 2 μM polymerase, 5 mM MgCl$_2$, 5 mM MnCl$_2$, 4 μM 5′ vRNA (5′-UAGUAGUAGUAUGCUCC-3′), 3′ vRNA (5′-UUUAGGGAGUCUACUACUA-3′), 10 μM 12mer capped RNA (5′ m$^7$GpppA-mAUCUAUAAUAG 3′), nucleotide triphosphates (ATP, GTP, UTP, CTP) to 1 mM and incubated at 30 ˚C for 60 minutes. The NaCl concentration in the sample was decreased to 250 mM. The sample was then pegylated with MS(PEG)8 Methyl-PEG-NHS-Ester (ThermoFisher Scientific) for 15 minutes prior to grid preparation. For samples used in dataset 3 and 4 an increased temperature was used to increase polymerase turnover compared to those samples prepared on ice.

All grids were prepared using a Vitrobot mark IV (FEI) at 100% humidity. Quantifoil Holey Carbon (R2/1, 200 mesh copper or R2/1 200 mesh gold) grids were glow discharged, before a volume of 3.5 μL sample at 0.5 mg/ml was applied and blotted for 3.5 seconds before vitrification in liquid ethane.

### Cryo-EM image collection and data processing

Cryo-EM data for all the bunyavirus polymerase sample were collected at the Oxford Particle Imaging Centre (OPIC), on a 300 kV G3i Titan Krios microscope (Thermo Fisher Scientific) fitted with a SelectrisX energy filter and Falcon IV direct electron detector. Automated data collection was setup in EPU 3.1–3.3 and movies were recorded in eer format. Sample were collected using Aberration-free image shift (AFIS) with a total dose of ~ 50 e-/Å$^2$, a calibrated pixel size of 0.932 Å/pix and a 10 eV slit. Sample-specific data collection parameters are summarized in S1 Table.

### Cryo-EM data processing

All datasets were processed using the cryosparc V4.0–4.2, following the same initial workflow. The eer format movies were fractionated in 60 frames without applying an up-sampling factor. Pre-processing was performed using patch motion correction and patch-CTF estimation with default settings. Corrected micrographs with poor statistics where manually curated. A first round of blob picking on the full dataset was followed by a round of 2D classification to generate initial templates. These templates were used for template picking on the full dataset. After 2D classification, the good-looking classes were selected. After this step different processing steps led to the reconstructions and are described for each different dataset.

For the HTNV-Apo dataset three ab-initio models were generated and further refined using heterogenous refinement. Particles belonging to the good-looking maps were used to train topaz model [37] and pick a new set of particles. These particles were 2D classified and the selected particles were used to generate three new ab-initio models that were then refined using heterogenous refinement. To recover rare view that could have been classified out during 2D classification, all the particles picked from blob picker, template picker and topaz picker directly 3D classified using heterogenous refinement with all the models generated with ab-initio. The previously good-looking maps were imported twice to help classifying out similar but different conformations. To remove duplicates, particles with an origin closer than 60 Å from each other were discarded. Two good classes were present, one presenting only the polymerase core and the other presenting an extra density corresponding to the endonuclease domain. Each of the models were refined using NU-refinement with the per particles CTF refinement option, leading to a final reconstruction of 2.78 Å and 3.24 Å respectively.

For the HTNV full length, the selected particles were used to train topaz model and pick a new set of particles. These particles were 2D classified and the selected particles as well as the particles selected from blob picker and template picker were pooled together. Particles with an origin closer than 60 Å from each other were discarded. Remaining particles were classified with a heterogenous refinement using the ab-initio models generated in the HTNV-Apo dataset. Out four good classes, two different conformations were identified and further processed separately. The three classes representing the core only were merged and refined using NU-refinement with the per particles defocus refinement option. Particles were then exported to Relion 3.1.3 for a round of 3D classification without alignment, 50 iterations and T = 4. After visual inspection in chimera, particles belonging to the interesting conformation were re-imported back into cryosparc and refined using local refinement to a resolution of 3.23 Å. The second conformation showing the core plus some extra domains was refined using NU-refinement. To get more particles representing this conformation, the particles belonging to that class were used to train a new topaz model and pick a new set of particles that were directly 3D classified using the previous classes from heterogenous refinement. One class showed extra density outside the core. Particles were then exported to Relion 3.1.3 for a round of 3D classification without alignment, 50 iterations and T = 4. After visual inspection in chimera, particles belonging to the interesting conformation were re-imported back into cryosparc and refined using local refinement to a resolution of 3.32 Å.

For the HTNV—core—5′ vRNA–NTP—Mg, the selected particles were used to train topaz model and pick a new set of particles. These particles were 2D classified and the selected particles as well as the particles selected from blob picker and template picker were pooled together. Particles with an origin closer than 60 Å from each other were discarded. Remaining particles were used to generate four ab-initio models followed by a heterogenous refinement. After visual inspection in chimera, particles belonging to the interesting conformation were refined using NU-refinement with the per particles CTF refinement option, leading to a final reconstruction of 3.23 Å.

For the HTNV—core—5′ vRNA, the selected particles were used to train topaz model and pick a new set of particles. These particles were 2D classified and the selected particles as well as the particles selected from blob picker and template picker were pooled together. Particles with an origin closer than 60 Å from each other were discarded. Remaining particles were classified with a heterogenous refinement using the ab-initio models generated in the HTNV-Apo dataset. The two good classes representing the core only were merged and refined using NU-refinement with the per particles defocus refinement option. Particles were then exported to Relion 3.1.3 for a round of 3D classification without alignment, 50 iterations and T = 4. After visual inspection in chimera, particles belonging to the interesting conformation were re-imported into cryosparc and refined using local refinement to a resolution of 2.79 Å.

To aid in placement of lower resolution domains deepEMhancer [38] post processing maps were generated to aid in the positioning of some domains. Where these have been used they have been uploaded as additional maps to the EMDB. Final model refinements were all performed against the original experimental maps.

## Structure determination and model refinement

The core of the HTNV-Apo (PDB ID: 8P1J) was first modelled by first fitting the individual domains predicted by Alphafold2 implemented in CollabFold [39,40] using UCSF ChimeraX [41]. Where the resolution of mobile domains (the endonuclease and/or C-terminal domains) was not sufficient to build ab initio, we utilised a hybrid modelling approach. For 8P1K we used our core structure (PDB ID: 8P1J) combined with existing Hantaan virus endonuclease

experimental structures (PDB ID: 5IZE) to guide the rigid-body fitting with minor manual adjustments. For the full-length model (PDB ID: 8P1L) both Alphafold2 and PDB ID: 5IZE were used to guide rigid-body fitting and refinement. In WinCoot 0.9.6, the restraints module was used to generate restraints at 4.3 Å and allow flexible refinement to fit the main chain into density. Multiple cycles of manual adjustment in WinCoot followed by real refinement in PHENIX were used to improve model geometry. The final model geometry and map-to-model comparison was validated using PHENIX MolProbity. The same approach has been used for all the structures but using HTNV-Apo model as an initial model and the All map and model statistics are detailed in S1 Table. Structural analysis and figures were prepared using UCSF ChimeraX [41].

## Supporting information

**S1 Table. Cryo-EM collection and refinement parameters.**
(XLSX)

**S1 Fig. Endonuclease activity of the HTNV polymerase.** (A) SDS-page analysis of the wt and D97A HTNV viral polymerase preparations. (B) Endonuclease activity assays in the presence or absence of divalent cations at fixed concentrations alone or in combination. (C) Titration of $MnCl_2$ to determine the effect on endonuclease activity. (D) Addition of viral RNA promoters to the endonuclease assay did not affect activity.
(TIF)

**S2 Fig. Cryo-EM processing workflow.** Processing schemes for the RNA free (A) and 5′ RNA only (B) HTNV cryo-EM datasets.
(TIF)

**S3 Fig. The polymerase core and RNA binding to the polymerase RdRp.** (A) Insertions into the palm domain from HTNV (grey) with insertions 1 (yellow) and insertion 2 (red) annotated. The structure of the LACV polymerase (PDBID: 6Z6G) (blue, green, pink) are shown with the California like insertion (purple). (B) The position of the prime-realign (PR) loop is shown in the retracted position from HTNV, DBV (PDBID: 6Y6K), LACV pre-initiation state (PDBID: 6Z6G), and RNA free Lassa virus (PDBID: 6KLC) polymerase are shown. The location of the PR loop and motif F is annotated. (C) Electron density for the bases observed in the 5′ hook structure from the 5′ vRNA bound + NTP model. (D) The HTNV vRNA binding site from the 5′ vRNA for HTNV (green) is compared to the 1–25 vRNA bound HTNV model (purple). (E) The alignment of the register of the 5′ vRNA from HTNV (green), LACV (blue, PDBID: 7ORK) and lassa virus (purple, PDBID: 7OJL) are shown with base numbers annotated. (F) The position of motif F changes in response to the binding of the 5′ vRNA promoter. (G) LACV polymerase (PDBID: 7ORK), when the 5′ vRNA promoter is bound, motif F is similarly arranged to that observed in HTNV. HTNV polymerase (green) and the LACV (yellow, red) are shown.
(TIF)

**S4 Fig. Cryo-EM processing workflow.** Processing schemes for the Full-length (A) and RNA and NTP (B) bound HTNV cryo-EM datasets.
(TIF)

**S5 Fig.** (A) Density map for the nucleotide in the polymerase active site with interactions residues annotated. (B) HTNV (grey) and DBV (green) (PDBID: 8ASD) polymerase are overlaid showing the conservation of the nucleotide binding site in the polymerase active site. The product and template RNA strands from DBV polymerase are shown in stick representation.

(C) Density map for residues 1167–1185 which undergo large rearrangement.
(TIF)

**S6 Fig.** Comparison of HTNV (PDBID: 8P1N) (A) and Sin Nombre (PDBID:8CI5) (B) core with 5' promoter RNA bound with domains annotated. (C) The HTNV structure has been coloured according to the RMSD of equivalent Ca. Global RMSD of equivalent (1022 residues) Ca is 1.03 Å.
(TIF)

## Acknowledgments

We thank members of the Grimes and Fodor laboratories for helpful comments and discussions. Access to electron microscopes was provided by the OPIC Electron Microscopy Facility (funded by Wellcome JIF (060208/Z/00/Z) and equipment (093305/Z/10/Z) grants). Access to computational resources was supported by a Wellcome Trust Core Award Grant (203141/Z/16/Z).

## Author Contributions

**Conceptualization:** Jeremy R. Keown, Loïc Carrique, Benjamin E. Nilsson-Payant, Jonathan M. Grimes.

**Data curation:** Jeremy R. Keown, Loïc Carrique.

**Formal analysis:** Jeremy R. Keown, Loïc Carrique, Benjamin E. Nilsson-Payant, Ervin Fodor, Jonathan M. Grimes.

**Funding acquisition:** Jeremy R. Keown, Benjamin E. Nilsson-Payant, Ervin Fodor, Jonathan M. Grimes.

**Investigation:** Jeremy R. Keown, Loïc Carrique, Benjamin E. Nilsson-Payant, Jonathan M. Grimes.

**Methodology:** Jeremy R. Keown, Loïc Carrique, Benjamin E. Nilsson-Payant.

**Project administration:** Jeremy R. Keown, Loïc Carrique, Benjamin E. Nilsson-Payant, Ervin Fodor, Jonathan M. Grimes.

**Resources:** Jeremy R. Keown, Benjamin E. Nilsson-Payant, Ervin Fodor.

**Software:** Jeremy R. Keown, Loïc Carrique.

**Supervision:** Ervin Fodor, Jonathan M. Grimes.

**Validation:** Jeremy R. Keown, Loïc Carrique, Benjamin E. Nilsson-Payant, Ervin Fodor.

**Visualization:** Jeremy R. Keown, Loïc Carrique, Benjamin E. Nilsson-Payant.

**Writing – original draft:** Jeremy R. Keown, Loïc Carrique, Benjamin E. Nilsson-Payant, Ervin Fodor, Jonathan M. Grimes.

**Writing – review & editing:** Jeremy R. Keown, Loïc Carrique, Benjamin E. Nilsson-Payant, Ervin Fodor, Jonathan M. Grimes.

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
