## [Decision Letter · Decision Letter 0]

29 Jul 2024

Dear Dr. Grimes,

Thank you very much for submitting your manuscript "Structural characterisation of the Hantaan virus polymerase describes a full-length polymerase conformation and reveals large rearrangements of the polymerase active site." for consideration at PLOS Pathogens. As with all papers reviewed by the journal, your manuscript was reviewed by members of the editorial board and by at least one independent reviewer. In your particular case,  we sought for multiple independent reviews but only manage to get one expert review. Instead of holding off on a  decision, we decided to proceed with the one review and a close look by the Section Editor.  In light of the review (below this email), we would like to invite the resubmission of a significantly-revised version that takes into account the reviewer's comments.

Pleas pay particular attention to how the novel features interpreted in this study can be determined by the resolution of cryoEM amps obtained. 

We cannot make any decision about publication until we have seen the revised manuscript and your response to the reviewers' comments. Your revised manuscript is also likely to be sent to reviewers for further evaluation.

Sincerely,

Benhur Lee

Section Editor

PLOS Pathogens

Benhur Lee

Section Editor

PLOS Pathogens

Michael Malim

Editor-in-Chief

PLOS Pathogens

orcid.org/0000-0002-7699-2064

Reviewer's Responses to Questions

**Part I - Summary**

Reviewer #1: The manuscript by Keown et al. describes several structures of the HTNV polymerase including protein constructs of two different lengths and with and without bound RNA. While significant structural work on HTNV polymerase has been previously published, this work details several new findings including conformations of motif F that block template binding and a full-length structure of the polymerase. Despite enthusiasm for these novelties, there are several concerns primarily relating to the interpretation of the authors’ EM maps. While the findings are interesting both the content and the descriptions in the text should be rigorously reevaluated.

**Part II – Major Issues: Key Experiments Required for Acceptance**

Reviewer #1: Line 260: The identification of a nucleotide in the active site is unjustified. Even taking the map threshold very low, it is not possible to distinguish any part of the nucleotide or magnesium ions from the noise. It isn’t clear how the authors produced figure S6A, but I cannot reproduce it with their maps.

Fig S7: Given the weak nature of the bands and large number of alternate products. I would prefer to see replicates of the ApG extensions with polymerase mutants.

**Part III – Minor Issues: Editorial and Data Presentation Modifications**

Reviewer #1: Line 133: Claims of an ordered endonuclease domain in 8P1K seem overstated. The resolution of this domain is significantly lower than the core portion of the structure and much of the domain is entirely absent at low threshold.

Line 137: Not clear what the reference to “261-273” is referring to as it appears to not be the endonuclease, core lobe or linker.

Line 158: the observation of insertions in the palm domain appear to make a claim to novelty, but these structures were previously observed in Durieux et al. 2023.

Line 189: Not clear what “different” piece of 5’ vRNA the authors are referring to here. 8QH3 appears to have an identical RNA sequence bound at this position.

Line 218: The C-terminal domain, while clearly present in the map, is largely disordered. While the domain as modeled is one possible interpretation of this map, structural homology searches and comparisons mentioned in the text are somewhat dubious.

Line 229: While there is clearly a shift in the endonuclease domain between 8P1K and 8P1L, the poor density for the endonuclease in map 17352 precludes any sort of conclusion about changes in this regions conformation or contacts as seen in the full-length structure.

Line 232: The authors have actually built three conformations for this loop. The density for this region in 8P1M/17354 is poor though and it isn’t sure what the justification for an alternate build here is.

Line 310: Presumably “the position” of motif E being referred to is the alpha helical conformation?

Line 325: I am not convinced that the authors are seeing four distinct conformations of motif E. 8P1J, 8P1K and 8P1M all have the same short helix, 8P1L contains the longer helix and 8P1N contains the beta-sheet. It isn’t clear here what the authors are referring to as the “ordered loop” arrangement.

Line 570: Update to indicate 8P1L pairing with EMDB-17353. 8P1M appears to be 5’ vRNA + ATP and 8P1N appears to bound to just 5’ vRNA. Database entries appear to be correct.

Fig 3A, 4ACD: Please avoid using yellow text, it is very hard to read.

Fig S2, S4: Figures and text need to be at a size that they can be read

PLOS authors have the option to publish the peer review history of their article (what does this mean?). If published, this will include your full peer review and any attached files.

Reviewer #1: No
---

## [Editor Report · Decision Letter 1]

20 Nov 2024

PPATHOGENS-D-24-00803R1Structural Characterization of the Full-length Hantaan Virus PolymerasePLOS Pathogens Dear Dr. Grimes, Thank you for submitting your manuscript to PLOS Pathogens. After careful consideration, we feel that it has merit but does not fully meet PLOS Pathogens's publication criteria as it currently stands. Therefore, we invite you to submit a revised version of the manuscript that addresses the points raised during the review process. Please submit your revised manuscript within 60 days Jan 19 2025 11:59PM. If you will need more time than this to complete your revisions, please reply to this message or contact the journal office at plospathogens@plos.org. Please include the following items when submitting your revised manuscript:*
A rebuttal letter that responds to each point raised by the editor and reviewer(s). You should upload this letter as a separate file labeled 'Response to Reviewers'. This file does not need to include responses to any formatting updates and technical items listed in the 'Journal Requirements' section below.*
A marked-up copy of your manuscript that highlights changes made to the original version. You should upload this as a separate file labeled 'Revised Manuscript with Track Changes'.*
An unmarked version of your revised paper without tracked changes. You should upload this as a separate file labeled 'Manuscript'. If you would like to make changes to your financial disclosure, competing interests statement, or data availability statement, please make these updates within the submission form at the time of resubmission. Guidelines for resubmitting your figure files are available below the reviewer comments at the end of this letter. We look forward to receiving your revised manuscript. Kind regards, Matthias Johannes Schnell, PhDSection EditorPLOS Pathogens Benhur LeeSection EditorPLOS Pathogens Michael Malim

Editor-in-Chief

PLOS Pathogens

orcid.org/0000-0002-7699-2064  **Journal Requirements:**

Please amend your detailed Financial Disclosure statement. This is published with the article. It must therefore be completed in full sentences and contain the exact wording you wish to be published.

Please ensure that the funders and grant numbers match between the Financial Disclosure field and the Funding Information tab in your submission form. Note that the funders must be provided in the same order in both places as well.

 **Reviewers' Comments:** **Figure resubmission:** While revising your submission, please upload your figure files to the Preflight Analysis and Conversion Engine (PACE) digital diagnostic tool, https://pacev2.apexcovantage.com/. PACE helps ensure that figures meet PLOS requirements. To use PACE, you must first register as a user. Registration is free. Then, login and navigate to the UPLOAD tab, where you will find detailed instructions on how to use the tool. If you encounter any issues or have any questions when using PACE, please email PLOS at figures@plos.org. Please note that Supporting Information files do not need this step. If there are other versions of figure files still present in your submission file inventory at resubmission, please replace them with the PACE-processed versions. **Reproducibility:** To enhance the reproducibility of your results, we recommend that authors of applicable studies deposit laboratory protocols in protocols.io, where a protocol can be assigned its own identifier (DOI) such that it can be cited independently in the future. Additionally, PLOS ONE offers an option to publish peer-reviewed clinical study protocols. Read more information on sharing protocols at https://plos.org/protocols?utm_medium=editorial-email&utm_source=authorletters&utm_campaign=protocols

---

## [Editor Report · Decision Letter 2]

26 Nov 2024

Dear Dr. Grimes,

We are pleased to inform you that your manuscript 'Structural Characterization of the Full-length Hantaan Virus Polymerase' has been provisionally accepted for publication in PLOS Pathogens.

Best regards,

Matthias Johannes Schnell, PhD

Section Editor

PLOS Pathogens

Benhur Lee

Section Editor

PLOS Pathogens

Michael Malim

Editor-in-Chief

PLOS Pathogens

orcid.org/0000-0002-7699-2064
---

## [Editor Report · Acceptance letter]

3 Dec 2024

Dear Prof. Grimes,

We are delighted to inform you that your manuscript, "Structural Characterization of the Full-length Hantaan Virus Polymerase," has been formally accepted for publication in PLOS Pathogens.

Best regards,

Sumita Bhaduri-McIntosh

Editor-in-Chief

PLOS Pathogens

orcid.org/0000-0003-2946-9497

Michael Malim

Editor-in-Chief

PLOS Pathogens

orcid.org/0000-0002-7699-2064